# Endotypes of Nasal Polyps in Children: A Multidisciplinary Approach

**DOI:** 10.3390/jpm13050707

**Published:** 2023-04-23

**Authors:** Emanuela Sitzia, Sara Santarsiero, Giulia Marini, Fabio Majo, Marcello De Vincentiis, Giovanni Cristalli, Maria Cristina Artesani, Alessandro Giovanni Fiocchi

**Affiliations:** 1Department of Otorhinolaryngology, Children’s Hospital “Ospedale Pediatrico Bambino Gesù—IRCCS”, 00165 Rome, Italy; emanuela.sitzia@opbg.net (E.S.); giulia.marini@opbg.net (G.M.); giovanni.cristalli@opbg.net (G.C.); 2Department of Cystic Fibrosis, Children’s Hospital “Ospedale Pediatrico Bambino Gesù—IRCCS”, 00165 Rome, Italy; fabio.majo@opbg.net; 3Faculty of Medicine and Surgery, University “Università Degli Studi Tor Vergata di Roma”, 00133 Rome, Italy; marcellodv1@libero.it; 4Department of Allergology, Children’s Hospital “Ospedale Pediatrico Bambino Gesù—IRCCS”, 00165 Rome, Italy; mariac.artesani@opbg.net (M.C.A.); agiovanni.fiocchi@opbg.net (A.G.F.)

**Keywords:** nasal polyps, children, chronic rhinosinusitis, CFTR-related disorder, antro-choanal polyp, primary ciliary dyskinesia, immunodeficiency

## Abstract

Nasal polyps (NPs) are rarely reported in childhood and usually represent red flags for systemic diseases, such as cystic fibrosis (CF), primary ciliary dyskinesia (PCD) and immunodeficiencies. The European Position Paper released in 2020 (EPOS 2020) provided a detailed classification and defined the correct diagnostic and therapeutic approaches. We report a one-year experience of a multidisciplinary team, made up of otorhinolaryngologists, allergists, pediatricians, pneumologists and geneticists, with the aim of ensuring a personalized diagnostic and therapeutic management of the pathology. In 16 months of activity, 53 patients were admitted (25 children with chronic rhinosinusitis with polyposis and 28 with antro-choanal polyp). All patients underwent phenotypic and endo-typic assessment, using proper classification tools for nasal pathology (both endoscopic and radiological), as well as adequate cytological definition. An immuno–allergic evaluation was carried out. Pneumologists evaluated any lower airway respiratory disease. Genetic investigations concluded the diagnostic investigation. Our experience enhanced the complexity of children’s NPs. A multidisciplinary assessment is mandatory for a targeted diagnostic and therapeutic pathway.

## 1. Introduction

Nasal Polyps (NPs) are rare in pediatric age. Compared to adults, in which the prevalence is between 1 and 4% of the general population, in children younger than 10 years old the frequency is estimated at 0.1% [1]. However, many cases of adult NPs might represent the manifestation of a process likely to have started in childhood or adolescence. Epidemiological data in pediatric age are scarce and most of the published studies refer to children with underlying systemic diseases [2].

The clinical suspicion of NPs arises when the patient complains of persistent and worsening nasal obstruction and smell reduction or loss, poorly responsive to medical therapies. NPs can be bilateral or unilateral. In children, differential diagnosis includes midline congenital defects, such as dermoid cyst, glioma and meningoencephalocele, as well as benign tumors, such as neurofibroma, craniopharyngioma and juvenile nasopharyngeal angiofibroma. Clinical, radiological and histopathological characteristics permit the exclusion of possible malignant forms that, although rare, can occur [3].

From an anatomic point of view, NPs in children can be distinguished in two different pathologic entities: chronic rhinosinusitis with nasal polyps (CRSwNP) and antro-choanal polyp (ACP) [2,4].

CRSwNP is one major phenotype of chronic rhinosinusitis (CRS). The prevalence of CRS in pediatric patients is now estimated to be up to 4% [5]. The emerging view is that CRS is a heterogeneous syndrome with a multifactorial etiology, where a dysfunctional interaction between environmental factors and the host’s immune system produces a well-defined chronic inflammatory pattern [6]. A bacterial infection is almost invariably present at the onset of the disease, but the mechanism that leads to chronicity is not always clear. Mirroring such studies, current guidelines give particular emphasis in pediatric age to secondary CRS, including cystic fibrosis (CF), primary ciliary dyskinesia (PCD), immunodeficiencies (ID) and vasculitis associated with Antineutrophil Cytoplasmic Antibodies (ANCA) [2,6]. CF remains the main cause of CRSwNP in children. PCD, in its various clinical expressions, often presents with chronic nasal sinus pathology and causes severe respiratory compromise over time resulting in reduced lung function. However, patients with CF, PCD and ID make up less than 20% of the CRS population [1,2].

More consistently, allergic inflammation has been indicated as a significant factor related to the morbidity of CRSwNP [2,7]. The prevalence of sensitization to environmental allergens is high in populations of children with CRSwNP [2,7]. In adults, allergy negatively impacts on patients’ quality of life (QoL) and CRSwNP control [8]. Despite this, the literature is not conclusive on the contribution of allergy to CRSwNP and the medical therapies used for allergic rhinitis treatment do not show a clear impact on CRSwNP prognosis [2]. Another pathogenetic mechanism often invoked in the case of CRSwNP is aspirin sensitivity [9]. Aspirin-exacerbated respiratory disease (AERD) is considered a typical condition of adults, but recently the emergence of pediatric cases leads to suspicion of a large underestimation in that age group [9].

The treatment of CRSwNP is traditionally based on medical procedures, including systemic and local corticosteroids and endoscopic sinus surgery (ESS) [2]. In recent years, endo-typic definition offered a new medical solution by targeting biological therapies for patients with CRSwNP [2,10]. In this context, CRSwNP is now interpreted as expression of an eosinophil-dominant inflammatory response, opposed to the neutrophilic nature of CRSsNP [11,12]. However, the similarities in pathophysiology among different clinical phenotypes has drawn attention to endotypes, subdivided based on pathophysiological mechanisms rather than phenotypes [13]. These endotypes have been classified, according to the expression of inflammatory mediators, into type 1, type 2 and type 3. Types 1 and 3 present neutrophils as cellular mediators, whereas type 2 CRS is characterized by an eosinophilic response, with an increased inflammatory pattern and high recurrence rate. Eosinophils are important for the remodeling process of nasal mucosa leading to NP pathogenesis, though other inflammatory pattern are involved [2,6].

ACPs, also known as Killian polyps, are benign masses that arise from the mucosa of the maxillary sinus, grow and reach the choana. ACPs account for approximately 4–6% of all NPs in the general population, increasing to 35% in children [14]. In contrast with CRSwNP, ACPs are typically unilateral, with nasal obstruction as the main symptom. The etiopathology of ACPs is poorly understood. In the past, mechanistic theories speculated that ACPs arise from an antral cyst, or a lymphatic obstruction, closing the osteo-meatal complex of the maxillary sinus, as a result of a chronic inflammation; increasing the pressure in the Highmoro antrum may force the herniation of the polyp into the nasal cavity through the accessory ostium [15]. Up to now, surgical removal is the only treatment option available for ACPs, and ESS is the most commonly recommended technique [16].

Current pathogenetic theories are unclear in explaining the high recurrence rate occurring in some patients and the development of ACPs before or during CRSwNP onset observed in some cases [17]. In this context, caseloads of CRSwNP in children are seldom reported [18,19], but could contribute to the understanding of the early origins of CRSwNP. In such instances, allergic cases were found to be more common than inflammatory cases in children when compared to adults [20].

Given these premises, we have set up a CRS clinic at our institution including the coordinated activity of otorhinolaryngologists, pulmonologists, pediatricians, allergists and cystic fibrosis experts. We present here a case series of children affected by NPs, either CRSwNP or ACP, attending the multidisciplinary Chronic Rhinosinusitis Allergo-Rhinologic Team (CRART) of our hospital.

## 2. Materials and Methods

Starting on 1 September 2021, CRART was established as a multi-disciplinary service aiming to offer complete diagnostic and therapeutic assistance for pediatric patients suffering from CRS. The service is articulated on the simultaneous presence of an allergist and an otolaryngologist in the clinical evaluation of the patient at all the various stages of the disease, with the availability of a cystic fibrosis expert and a pulmonologist for children affected by the respective pathologies. The service also makes use of a pathologist for histological and a radiologist for imaging evaluations. CRART delivers periodic evaluation of the following aspects: phenotypic and endo-typic identification of patients with CRS; clinical classification of nasal pathology (endoscopic and radiological signs); nasal cytology; histopathological definition (when undergoing surgery); and tissue bank activation. We periodically evaluate nasal microbiome, radiologic imaging, allergy and immune status, spirometry and quality of life (QoL). The service is intended for patients aged 2–18 years with CRS, with or without nasal polyps, including those with antro-choanal polyps and those with CF.

Per CRART protocol, all data on patients’ general medical history, including family, physiological, employment, remote pathology, recent pathology and pharmacological, was collected. CRS symptoms, including nasal obstruction or congestion, discharge or rhinorrhea, facial pain, headache, smell reduction or loss and cough were registered; allergic rhinitis symptoms were also recorded, including nasal congestion, discharge, nasal or eye itching and sneezing.

We assessed patients’ perceived quality of life (QoL) correlated with their nasal pathology by validated QoL questionnaire: the Sino Nasal Outcome Test 22 (SNOT-22) for patients 12 years of age or older and the Sino and Nasal Quality of life Survey (SN-5) for children under 12 years of age [21,22]. The ear-nose-throat (ENT) specialists performed basic objective examination (otoscopy, pharyngoscopy, anterior rhinoscopy) and nasal endoscopy by flexible fiberscope. A standardized scale, the Lund-Kennedy score, was used for endoscopic assessment [23]. In children, adenoidal hypertrophy was evaluated in grades from I to IV [24]. When already performed, nasal and sinus computed tomography (CT) imaging were evaluated using the standardized Lund-Mackay score [25].

A periodic evaluation of the respiratory function was carried out using a calibrated spirometer (Cosmed Micro Quark, Albano Laziale, 000141, Italy). For nasal cytology, samples taken from the middle third of the inferior turbinate were prepared for fresh evaluation of the ciliary motility and subsequently mounted with May Grunwald-Giemsa staining for evaluation of the rhino-cytogram [26]. Nasal swabs for investigations of the nasal microbiome (Isohelix DNA swab, Cell Projects Ltd., Maidstone, Kent, UK) were taken in the area of the middle nasal meatus, placed on ice upon collection and frozen within an hour at −80 °C until DNA extraction [27].

We evaluated sensitization status for inhalants at skin prick test (SPT) with respiratory allergens (Lofarma, Milano, Italy), as described elsewhere [28] and and/or specific IgE determination (ImmunoCAP^®^, Thermo Fisher Scientific, Uppsala, Sweden). Specific blood tests for immunodeficiency and vasculitis were also performed, including IgA and IgM level, total IgG and IgG subclasses level, and specific dosage of tetanus, diphtheria and pneumococcal IgG, ANCA test (c-ANCA and p-ANCA levels). We recorded the results of the sweat test and molecular analysis of CFTR and PCD genes.

This retrospective case series study focuses on children affected by CRSwNP attending CRART from September 2021 to December 2022. The local Ethics Committee authorized the review of the clinical records of children, corresponding to our inclusion criteria (13 December 2022, prot. 1512/22). Specifically, patients’ clinical phenotypes were classified based on the EPOS 2020 guidelines [2]. On the grounds of nasal endoscopy and CT imaging, patients were divided into two main groups: CRSwNP and ACP. To this end, we derived data on endo-typic immuno-profiling from patients’ medical records and classified patients into non-type2 or type2 according to EPOS 2020′s cut-off levels (tissue eos ≥ 10/hpf, OR blood eos ≥ 250, OR total IgE ≥ 100) [2]. Microsoft Excel was used for data collection and analysis.

## 3. Results

A total of 53 children were evaluated. Twenty-five patients were affected by CRSwNP, 12 females and 13 males, with a median age of 12 years (range 5–18 years). Twenty-eight children were diagnosed with ACP, 11 females and 17 males, with a median age of 12.5 years (range 6–18 years). The caseload characteristics are reported in Table 1.

### 3.1. CRSwNP Patients

Endotype 2 was identified in 10 children, four females and six males, with a median age of 12 years old (range 9–18 years old) (see Table 2). In particular, 5/10 were ascribed to such an endotype for tissual eosinophilia (tissue eos ≥ 10/hpf), 7/10 presented total IgE > 100 and 2/10 patients presented tissual eosinophilia and total IgE > 100. Mean total IgE was 243.57 ± 196.06, with a median of 176 (range 101–698). No patients presented blood eosinophilia ≥ 250. All such patients presented neutrophils in their rhino-cytogram, while eosinophils were observed in nine patients and mast cells in three. Two patients with positive sweat test were affected by CF. The incidence of inhalant allergic sensitization was 50% (with *Dermatophagoides* spp. sensitization in 5/10, alternaria alternata in 2/10, grass pollen in 3/10, cypress in 2/10, olive in 2/10 and cat dander in 3/10). One half of these cases did not present with any involvement of the lower airways, two suffered from asthma (both were allergic to several inhalant allergens), two CF patients presented bronchi-ectasias and one presented chronic bronchitis.

Fifteen patients were non-endotype 2, eight females and seven males, with a median age of 11 years old (range 5–18 years old) (see Table 3). One patient did not present inflammatory cells in the rhino-citogram. Neutrophils were detected in 14 cases, eosinophils in eight cases and mast cells in three cases. Among the six patients with positive sweat test results, five children were known to be affected by CF and one male was not (genetic characterization in progress). Four patients were known to be affected by PCD and one was affected by RAG1 deficiency. Lower airway pathology was detected in all these patients. In this group of patients, inhalant sensitization was remarkably rarer, with only four cases of inhalant sensitization (to *Dermatophagoides* spp. in 2/4, alternaria alternata in 1/4, grass pollen in 1/4, other pollens in 2/4, and cat/dog dander, respectively, in 2/4 and 1/4). The chi-square for allergic sensitization in endotype 2 vs. non-endotype 2 is 2.9231 (p.087321).

ANCA test proved negative in both groups. None presented with a history of allergy to aspirin or non-steroidal anti-inflammatory drugs.

### 3.2. ACPs Patients

Table 4 summarizes the clinical features of 28 patients with ACP, 17 males and 11 females, from 6 to 18 years old (mean age 12.85, SD = 3.29, median age 12.5). Allergic sensitization was present in eight patients (28.57%), with *Dermatophagoides* spp. sensitization in 7/8, alternaria alternata in 1/8, grass pollen in 3/8, cypress in 1/8, olive in 2/8 and cat dander in 2/8 and dog dander in 1/8. The rhyno-cytogram showed inflammatory cells in all patients’ samples. Neutrophils were detected in 27 patients, alone (10 patients) or associated with eosinophils (13 patients), or eosinophils and mast cells (four patients), while one patient presented eosinophils alone. Immuno-profiling showed an endotype 2 inflammation in seven patients (25%), with 4/7 ascribed to tissual eosinophilia (tissue eos ≥ 10/hpf), 1/7 presented total IgE > 100 and 2/7 presented tissual eosinophilia and total IgE > 100 and 1/7 (mean total IgE 227.66 ± 32.78, median total IgE = 236, range 184–263). No patients presented with blood eosinophilia ≥ 250. Lower airways disease (asthma) was detected in two patients, one with allergic sensitization and one with intrinsic asthma. One non-endotype 2 female was diagnosed with IgA deficiency. ANCA tests proved inconclusive for all patients.

## 4. Discussion

There are very few caseloads of NPs in children. In this study, we wanted to set the problem in a global manner, starting from the clinical manifestation, to describe all the possible underlying pathogenesis. For this reason, we did not exclude children whose general diagnosis had already been made (so we also included children with CF and other chronic respiratory diseases, CRD). In this peculiar approach, consistent with the clinical reality, pediatric NPs emerges as a composite condition.

Compared to Italian adult caseloads, our series of CRSwNP patients presents a lower prevalence of allergic sensitization (26% vs. 50.8%) [30]. They express a Th2 endotype with less frequency (35.7 vs. 76.8%), do not present with a personal history (none vs. 24.8%) and display a low frequency of family history of allergy to non-steroidal anti-inflammatory drugs (AERD). From our point of view, therefore, the disease seems much more multifaceted than it is in adult patients. In adults with CRSwNP, immuno-profiling is the new frontier for diagnostic–therapeutic management. Since 2019, the US Food and Drug Administration (FDA) and European Medicines Agency (EMA) have approved dupilumab, an IL-4Rα inhibitor, in the treatment of type 2 CRSwNP [10]. As guidelines suggest, non-responder adults should be evaluated for secondary CRSwNP [2]. Pending genetic evaluations, our results indicate that, in pediatric patients, an opposite approach could be appropriate, as clinicians should consider a secondary CRSwNP highly probable. Early diagnosis of secondary CRSwNP is mandatory for preventing evolution of nasal pathology and for identifying other possible organs involved in the disease, radically changing the therapeutic approaches and prognosis.

In this context, a lower airways disease state assessment is crucial. In our study, almost all the children with CRSwNP are affected by several lower airway pathologies, including chronic bronchitis, recurrent pneumonia, bronchiectasis, asthma and reactive airway disease (RAD). A strong cooperation between ENT and pulmonologist is appropriate, in order to apply the principles of united airway disease (UAD) in diagnostic and therapeutic management.

The prevalence of allergic sensitization in our caseload of CRSwNP (36%) did not exceed the prevalence of skin prick test positivity in open populations of Central-Southern Italian adolescents (41%) [31]. This rather surprising finding further differentiates the epidemiology of children with CRSwNP from that of adults.

Nevertheless, allergists have an important role in patients’ assessment because, even though not being a causative agent, allergic rhinitis (AR) and asthma may co-exist in NPs children and can impact negatively on global airway inflammation and patients’ QoL [32]. Indeed, allergic inflammation can overlap with other inflammatory pathways and predisposing conditions in the individual, thus contributing to CRSwNP pathophysiology. Allergy is characterized by type-2 helper T-cell (Th2) cytokine-mediated inflammation in the nasal mucosa, sharing a similar inflammatory pathway with endotype-2 CRS [8]. Polyclonal IgE has been identified in the polyp tissue, in allergic as well as in non-allergic patients with CRSwNP, therefore local IgE production can result from different inflammatory pathways [33]. In adults, a major role has been attributed to persistent allergic rhinitis, in particular house dust mite (HDM) sensitization, although polysensitization generally worsens nasal inflammation and treatment response [34]. All our allergic children are sensitized to dust mites and a significant percentage are poly-allergic. Reducing allergic inflammation improves CRSwNP prognosis [35]. Nasal or systemic corticosteroids are useful in allergic and non-allergic CRSwNP, thus they are recommended in both allergic rhinitis and CRSwNP alone.

Although the literature lacks strong evidence, a similar approach has been suggested in children with ACPs [36]. In our high recurrence ACPs, immuno-profiling suggests a non-type 2 endotype, confirming the data reported in the literature [37]. Thus, in children also, the role of allergy in ACPs pathogenesis is unclear [38].

We have reported data about nasal cytology, which is particularly useful in pediatric age, due to its low invasivity and simple execution. Sample collection and observation are replicable and comparable [39]. In our case series, all patients with NPs present inflammatory cells in their nasal mucosa. Neutrophils are the most commonly detected, showing that, in these patients, nasal mucosa suffers from persistent inflammation, regardless of the type of immunological or genetic profile, in both allergic and non-allergic. As expected, in type 2 patients, eosinophils were detected in various grade, with mast cells presenting in three cases. In non-type 2 patients, eosinophils were observed in four non allergic patients (three patients with CRSwNP and one patient with ACP), defining a non-allergic rhinitis with eosinophils (NARES). One non-endotype 2 CRSwNP non allergic patient had eosinophils and mast cells, defining a non-allergic rhinitis with mast cells (NARESMA). NARES and NARESMA are classified among the non-allergic or cellular forms of rhinitis (NAR), clinical entities identifiable only by a nasal cytology assessment. Rather than being simply descriptive, patient’s nasal cytology can guide clinicians in evaluating and in monitoring the nasal mucosa inflammatory state before, during and after medical treatment, or after surgery, even in children with NPs [40]. Eosinophil and mast cells detection represents a suggestive prognostic factor for CRSwNP development, while neutrophils’ persistence is a sentinel of chronic nasal inflammation and poor nasal disease control [41]. As known, neutrophils rather than eosinophils domain nasal inflammation in CF and ACP patients [42]. Nevertheless, eosinophils and mast cells can also be found in the nasal mucosa of these patients and make a variable contribution to nasal disease progression. Further studies are needed to better determine the potential role of nasal cytology in the management of NPs in children, e.g., monitoring nasal inflammation state during the administration of the new biologic therapies in CF patients [43].

In our case series, only one patient was affected by congenital immunodeficiency (RAG1 deficiency) and no ANCA-associated vasculitis emerged. Furthermore, only two patients (one with CRSwNP and one with ACP) have a familiar AERD and no patients still manifest personal AERD. All these are rare pathologies in the general population and, in childhood, first diagnoses are often performed before NPs’ development for the serious systemic conditions occurring at first stages of pathology. Nevertheless, onsets in late childhood or adolescence are possible, in particular for common variable immunodeficiency (CVID). In children with CRSwNP, immune testing is strongly suggested, particularly when middle ear or lower respiratory tract infections are associated (recurrent pneumonia, bronchiectasis) [44]. Whether the results are abnormal, or normal with a high suspicion, patients should be referred to a clinical immunologist.

The specific determination of cytokines, e.g., IL4, IL5 or IL13, better define the endotype and their quantification is useful for sub-classifying different levels of type 2 inflammation, as reported in a recent work [45]. Further studies will identify proper biomarkers that can hide behind clinically analogous phenotypes, in type 2 as well as non-type 2 patients.

In conclusion, pediatric NPs is a variable phenotype caused by a complex interaction of different factors, not all understood until now. A well-designed, multidisciplinary approach is mandatory for NPs management in children. The strong cooperation between different specialists is fundamental to arrive at personalized diagnostic tools and target therapies.

## 5. Conclusions

NPs is a rare condition in children but should be suspected and excluded in case of persistent and worsening nasal obstruction, either unilateral or bilateral. NPs at pediatric age should be studied in a multidisciplinary context in order to identify all the possible predisposing factors and comorbidities. Better knowledge of the pathogenesis of NPs in children could lead to identification of molecular biomarkers and the future development of precision therapies. Although NPs has different expressions in pediatric age compared to adult, the authors are convinced that the premises of this pathology are often present in childhood; investigating and recognizing these, as well as being good clinical practice, could provide better therapeutic perspectives.

## Figures and Tables

**Table 1 jpm-13-00707-t001:** Demographic and general characteristics of 53 pediatric patients with NPs.

	CRSwNP Patients	ACP Patients
*N*	25	28
M/F	13/12	17/11
Mean age, median age, range (y)	12.08, 12, 5–18 y	12.85, 12.5, 6–18 y
Sensitized to aeroallergens	9/25 (36%)	8/28 (28.57%)
Lower airway disease *	20/25(80%)	0/28 (0%)
AERD (personal)	0/25	0/28
AERD (familiar)	1/25	1/28

* Chronic bronchitis, bronchiectasis and/or asthma; AERD: Aspirin-exacerbated respiratory disease.

**Table 2 jpm-13-00707-t002:** Patients with CRSwNP and endotype 2.

Patient No, Age, Gender	Allergy	Nasal Cytology	Sweat Test	Lower Airway Disease
N = 1, 12 y, F	No	NEU3+, EOS1+	Positive	bronchiectasis
N = 2, 12 y, F	No	NEU4+, EOS1+	Positive	bronchiectasis
N = 3, 9 y, M	Yes	NEU 4+, EOS3+	negative	asthma
N = 4, 12 y, F	Yes	NEU3+, EOS2+, MAST1+	negative	asthma
N = 5, 13 y, M	Yes	NEU4+, EOS2+, MAST1+	borderline	none
N = 6, 9 y, F	No	NEU4+, MAST1+	negative	chronic bronchitis
N = 7, 9 y, M	Yes	NEU4+, EOS4+	negative	none
N = 8, 15 y, M	No	NEU4+, EOS3+	negative	none
N = 9, 14 y, M	Yes	NEU2+, EOS2+	negative	none
N = 10, 18 y, M	No	NEU4+, EOS2+	negative	none

N: patients’ identification number; NEU: neutrophils; EOS: eosinophils; MAST: mast-cells. Inflammatory cell count was performed by a semiquantitative, standardized method from Gelardi et al. [29]. In particular, cell count was defined as grade 0 (not visible), grade 1+ (occasional groups), grade 2+ (moderate number), grade 3+ (easily visible), grade 4+ (elevated number).

**Table 3 jpm-13-00707-t003:** Patients with CRSwNP and non-type 2 endotype.

Patient No, Age, Gender	Allergy	Nasal Cytology	Sweat Test	Lower Airway Disease
N = 11, 5 y, M	No	NEU1+, EOS1+	Positive	bronchiectasis
N = 12, 6 y, F	No	NEU1+, EOS1+, MAST1+	Positive	bronchiectasis
N = 13, 6, M	No	NEU3+	Positive	bronchiectasis
N = 14, 14 y, F	No	NEU2+	Positive	bronchiectasis
N = 14, 15 y, F	No	NEU1+	Positive	bronchiectasis
N = 16, 18 y, M	No	NEU3+, EOS1+	Positive	bronchiectasis
N = 17, 5 y, F	Yes	NEU4+, EOS1+, MAST3+	Negative	asthma
N = 18, 10 y, M	Yes	NEU3+, EOS1+	borderline	asthma
N = 19, 18 y, F	Yes	MAST1+	Negative	asthma
N = 20, 11 y, F	Yes	NEU4+, EOS1+	Negative	chronic bronchitis
N = 21, 11 y, M	No	NEU1+, EOS2+	Negative	bronchiectasis
N = 22, 13 y, M	No	NEU4+	Negative	bronchiectasis
N = 23, 18 y, M	No	NEU2+	Negative	bronchiectasis
N = 24, 11 y, F	No	NEU4+, EOS2+	Negative	reactive airway disease
N = 25, 18, M	No	NEU3+	Negative	bronchiectasis

N: patients’ identification number; NEU: neutrophils; EOS: eosinophils; MAST: mast-cells. Inflammatory cell count was performed by a semiquantitative, standardized method from Gelardi et al. [29]. In particular, cell count was defined as grade 0 (not visible), grade 1+ (occasional groups), grade 2+ (moderate number), grade 3+ (easily visible), grade 4+ (elevated number).

**Table 4 jpm-13-00707-t004:** Patients with ACPs.

Patient No, Age, Gender	Allergy	Nasal Cytology	Sweat Test	Lower Airway Disease	Endotype
N = 26, 9 yo, M	No	NEU4+	negative	none	Non type-2
N = 27, 9 yo, M	Yes	NEU2+, EOS1+	negative	none	Non type-2
N = 28, 18 yo, M	No	NEU4+, EOS1+	negative	none	Non type-2
N = 29, 13 yo, F	Yes	NEU4+	negative	none	Non type-2
N = 30, 16 yo, M	No	NEU3+	negative	none	Non type-2
N = 31, 17 yo, M	No	NEU2+	negative	none	Non type-2
N = 32, 6 yo, M	Yes	NEU1+/EOS1+	negative	none	Endotype 2
N = 33, 12 yo, M	No	NEU4+	negative	none	Non type-2
N = 34, 12 yo, M	No	NEU4+	negative	none	Non type-2
N = 35, 14 yo, F	Yes	NEU2+, EOS1+, MAST3+	negative	none	Non type-2
N = 36, 11 yo, M	No	NEU3+	negative	none	Non type-2
N = 37, 15 yo, F	No	NEU1+, EOS2+	negative	none	Endotype 2
N = 38, 16 yo, F	No	NEU3+	negative	none	Non type-2
N = 39, 9 yo, M	No	EOS1+	negative	none	Non type-2
N = 40, 18 yo, F	No	NEU1+	negative	none	Non-type 2
N = 41, 8 yo, F	No	NEU4+,EOS1+	negative	none	Endotype 2
N = 42, 10 yo, M	No	NEU3+/EOS1+	negative	none	Non-type 2
N = 43, 12 yo, M	No	NEU3+/EOS1+	negative	asthma	Non-type 2
N = 44, 15 yo, F	No	NEU1+	negative	none	Non-type 2
N = 45, 9 yo, F	No	NEU3+/EOS1+	negative	none	Non-type 2
N = 46, 18 yo, M	Yes	NEU3+/EOS1+	negative	none	Endotype 2
N = 47, 17 yo, M	No	NEU4+/EOS4+	negative	asthma	Endotype 2
N = 48, 10 yo, M	Yes	NEU3+, EOS2+, MAST1+	negative	none	Endotype 2
N = 49, 13 yo, M	No	NEU3+, EOS2+	negative	none	Endotype 2
N = 50, 12 yo, F	No	NEU4+, EOS3+, MAST1+	negative	none	Non-type 2
N = 51, 12 yo, F	No	NEU4+, EOS2+	negative	none	Non-type 2
N = 52, 14 yo, F	Yes	NEU2+, EOS1+	negative	none	Non-type 2
N = 53, 15 yo, M	Yes	NEU1+, EOS3+, MAST1+	negative	none	Non-type 2

N: patients’ identification number; NEU: neutrophils; EOS: eosinophils; MAST: mast-cells. Inflammatory cell count was performed by a semiquantitative, standardized method from Gelardi et al. [29]. In particular, cell count was defined as grade 0 (not visible), grade 1+ (occasional groups), grade 2+ (moderate number), grade 3+ (easily visible), grade 4+ (elevated number).

## Data Availability

Not applicable.

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
