# Peer review of "Endotypes of Nasal Polyps in Children: A Multidisciplinary Approach"

_jpm, 2023, doi:10.3390/jpm13050707_

Round 1

Reviewer 1 Report

This is a very interseting study which results demonstrating the need for multidisiplinary approach for treatment in nasal polyposis in the pediatric population.  This approach is mandatory  for adults as well whenever there is a presentation with multy -organ involevement (as asthma , reccurent infections , skin etc). 

It will be interesting to discuss why in the authors opinion there were no patients with AERD (is it because the young age? 

It will be interesting to follow these patients and treatments outcome according to their phenotypes.     

Author Response

Dear Reviewer, we are pleased to receive your interest for our study. We observed the absence of AERD patients in our case series. We agree with your opinion: a late manifestation of this pathology is possible. We will certainly follow the patients over time to evaluate evolutions in the clinical phenotype. In this regard, we have added a comment in the discussion section. Thanks for the contribution.

Best regards,

Sara Santarsiero, MD

Reviewer 2 Report

This is a retrospective case series study on the multidisciplinary approach in the endotyping of nasal polyposis in children.

The authors have made an interesting effort to collect the data from the CRSwNP and PCA, contextualizing different aspects that phenotype these patients and provide new perspectives in the future, from the diagnostic and therapeutic point of view.

They have found important similarities in the CRSwNP between adults and children, with respect to type 2 inflammation, and have also reported the neutrophilic predominance of PCA and its relative predisposition to suffer from concomitant secondary rhinosinusitis.

Tip #1. I suggest changing the term endotyped to the term phenotyping, since a determination of cytokines is not being carried out, but other descriptive aspects such as cellularity are being taken. Being strict, endotyping refers to the specific determination of, for example, IL4, IL5 or IL13.

Author Response

Dear Reviewer, we are pleased to receive your interest for our study. Regarding your suggestion (Tip#1), we used the term “endotype” because the patients were distinguished in “type 2” and “non-type 2” endotypes basing on the current definition accepted in international literature of “type 2 inflammation”, based on one of these EPOS criteria: blood, eosinophils >250 cells per microliter or blood total IgE >100 UI/L or local eosinophilia in the sinonasal mucosa with at least a mean of 10 cells/hpf at microscopic observation (Fokkens WJ, et al., European Position Paper on Rhinosinusitis and Nasal Polyps 2020. Rhinology. 2020 Feb 20;58 (Suppl S29):1-464. doi: 10.4193/Rhin20.600. PMID: 32077450). The specific determination of cytokines, e.g., IL4, IL5 or IL13, better define the endotype, and their quantification is useful for sub-classifying different levels of type 2 inflammation, as reported in a recent work published on JPM (De Corso, E.; Baroni, S.; Settimi, S.; Onori, M.E.; Mastrapasqua, R.F.; Troiani, E.; Moretti, G.; Lucchetti, D.; Corbò, M.; Montuori, C.; et al. Sinonasal Biomarkers Defining Type 2-High and Type 2-Low Inflammation in Chronic Rhinosinusitis with Nasal Polyps. J. Pers. Med. 2022, 12, 1251. https://doi.org/10.3390/ jpm12081251). In EPOS guidelines, phenotypes are clinical manifestation of nasal pathology, e.g., chronic rhinosinusitis with, or without nasal polyps (CRSwNP, or CRSsNP), antro-choanal polyps (ACPs), allergic fungal rhinosinusitis (AFRS), or nasal and systemic pathology, e.g., cystic fibrosis (CF), or primary ciliary dyskinesia (PCD), regardless of differences in molecular markers (Fokkens WJ, et al., European Position Paper on Rhinosinusitis and Nasal Polyps 2020. Rhinology. 2020 Feb 20;58(Suppl S29):1-464. doi: 10.4193/Rhin20.600. PMID: 32077450). We are already waiting for preliminary results on biomarkers to better define the different endotype that can hide behind clinically analogous phenotypes. In this regard, we have added a comment in the discussion section. Further studies will be published on this field by our study group. We thank you for the contribution.

Best regards,

Sara Santarsiero, MD
